# A retrospective two-center cohort study of the bidirectional relationship between depression and tinnitus-related distress

Cosima F. Lukas [1], Birgit Mazurek[2], Petra Brueggemann[2], Markus Junghöfer[3], Orlando Guntinas–Lichius [1] & Christian Dobel [1,4] ✉

## Abstract

**Background** Tinnitus can cause considerable psychological distress among patients, particularly if comorbidities occur. Despite a strong relationship between tinnitus-related distress and depression, the underlying mechanisms represent a long-standing question. By investigating the co-development of tinnitus-related distress and depressiveness throughout therapy, we capture the dynamic interplay of both conditions and uncover underlying common features mediating their link.

**Methods** Large datasets from two different day clinics in Germany have been analyzed using a regularization method for predictor selection (analysis 1) and latent growth curve modeling (LCM; analysis 2). Tinnitus-related distress was assessed using the Tinnitus Questionnaire (TQ). All patients have been experiencing chronic subjective tinnitus with a minimum mean severity level of TQ grade 2. Treatment at both day clinics involved tinnitus management according to clinical guidelines with minor idiosyncratic differences. Analysis 1 was performed on a dataset of 500 patients who received the Jena Interdisciplinary Treatment for Tinnitus (JITT) for 5 consecutive days between 2013 and 2017. Analysis 2 was performed on a second dataset, which included 1016 patients treated at the Tinnitus Center of the Charité Universitätsmedizin Berlin for 7 days between 2011 and 2015.

**Results** Here, we show a substantial bidirectional relationship between tinnitus-related distress and depression severity while emphasizing the role of somatic symptoms and perceived stress in the experience and maintenance of tinnitus awareness. The LCM provides adequate model fit (CFI = 0.993, SRMR = 0.016).

**Conclusions** Our results indicate enhanced therapy success in depression when tinnitus-related distress is addressed and vice versa. The combined treatment of tinnitus and depression is proposed for future treatment strategies.

## Plain language summary

Tinnitus, also described as ringing in the ears, can lead to considerable psychological distress and often occurs with depression. This study aimed to explore the relationship between tinnitus-related distress and depression. We analyzed data from two German day clinics to understand how tinnitus-related distress and depression interact during therapy. The main finding is a strong bidirectional relationship between tinnitus-related distress and depression. Physical complaints and stress explain part of this association. The study highlights the importance of addressing both tinnitus-related distress and depressive mood in a combined treatment. It suggests that reducing distress in one condition can enhance improvement in the other. This insight can make treatment better for individuals with chronic tinnitus and depression.

Subjective tinnitus is a common audiological phenomenon that leads to an internal perception of noise without any corresponding external source[1]. Chronic tinnitus, persisting after an acute phase of three months, causes a considerable individual and societal burden[2]. The lack of standardized assessment, geographical biases, and inconsistencies in defining and reporting tinnitus have led to high variability in prevalence estimates[3,4]. According to a 2021 pan-European epidemiological study, 14.7% of the general population report tinnitus, while 1.2% suffer from severely

bothersome tinnitus, which can be detrimental to emotional and social well-being and is associated with considerable psychological distress[5].

Tinnitus-related distress, the psychological reaction to chronic tinnitus, can be accompanied by concomitant psychological comorbidities such as anxiety, somatoform disorders, and, in particular, depression[6]. Therapeutic approaches, such as psychotherapeutic and sound-based interventions, aim at alleviating tinnitus-related distress as opposed to eradicating the tinnitus sound. As recommended by the current German S3 guideline

[1]Department of Otorhinolaryngology, Jena University Hospital, Friedrich-Schiller-University Jena, Jena, Germany. [2]Tinnitus Center, Charité Universitätsmedizin Berlin, Berlin, Germany. [3]Institute of Biomagnetism and Biosignalanalysis, University of Münster, Münster, Germany. [4]Department of Social and Behavioral Sciences, City University of Hong Kong, Hong Kong SAR, China. ✉e-mail: christian.dobel@med.uni-jena.de

for chronic tinnitus[7], cognitive behavior therapy (CBT) represents an effective treatment approach[8,9]. Despite the overall beneficial results, treatment responses have been characterized by large interindividual variations[10]. The heterogeneity in tinnitus perception, etiology, and related comorbidity hinders attempts at predicting tinnitus-related distress and treatment response[10]. Additionally, the application of rather basic statistical approaches and small sample sizes have been considered problematic in earlier studies[11].

Because psychoacoustic measures purporting to characterize the tinnitus sound, such as tinnitus loudness or pitch, cannot explain interindividual differences in the experience and maintenance of tinnitus-related distress, psychological features may be more strongly associated with tinnitus severity[12].

Various studies emphasize the unique role of depressive symptoms in tinnitus patients[6,13,14]. A global 2019 systematic review found a median 33%-prevalence of depression among tinnitus patients, substantially exceeding the prevalence of depression among the general population[14]. Both disorders have been shown to be pathophysiologically interrelated, suggesting a mutual interaction of tinnitus-related distress and depressiveness[13]. The continuous experience of tinnitus-related distress can induce long-lasting changes in depressive symptoms, while depression can contribute to the absence of emotional and attentional control under tinnitus-induced stress[15]. While the association of severe tinnitus-related distress and depression has been well-established in previous research[6,13,14], the underlying mechanisms of this relationship remain unclear. Somatization and symptoms of distress are discussed as possible mediating factors[13,16].

Bivariate latent growth curve models (LCMs) allow for estimating interindividual variability in intraindividual change processes over time in two latent constructs simultaneously[17]. This method is used here to investigate the dynamic interaction between tinnitus-related distress and depression severity at baseline and throughout their co-development over the course of treatment in a cohort of patients diagnosed with severe chronic subjective tinnitus, undergoing tinnitus-specific treatment at a specialized day clinic.

With the growing availability of high-dimensional datasets, methods derived from machine learning are incorporated more frequently into variable selection processes in clinical research[18]. To improve the accuracy of the LCMs acquired here, a regression-based machine learning method for automated predictor selection was performed on data from another cohort of patients with severe chronic subjective tinnitus who were treated at a different specialized Tinnitus-Center (analysis 1) in preparation for the LCM analysis (analysis 2).

Both datasets contained demographic, tinnitus-, and mental health-related data of patients diagnosed with severe chronic subjective tinnitus treated at the Tinnitus-Center of the Ear, Nose, and Throat (ENT) Department of the Jena University Hospital (JUH), Germany (analysis 1) and the Tinnitus-Center of the Charité Universitätsmedizin Berlin, Germany (analysis 2) respectively. Both day clinics employ multimodal treatment regimens that aim to achieve habituation and alleviate tinnitus-related distress using evidence-based interventions following the German S3 guideline for chronic tinnitus[7] with minor idiosyncratic differences (see methods section for a detailed description of treatment regimens and inclusion criteria). All patients had been suffering from clinically confirmed chronic subjective tinnitus aurium as diagnosed by an ENT specialist. The severity of corresponding tinnitus-related distress has been evaluated by a clinical psychologist.

With the primary aim of investigating the dynamic interaction between tinnitus-related distress and depression severity, a preparatory predictor selection is performed on one dataset. The retained features are reinvestigated using LCM within the second dataset to analyze the underlying mechanisms of this relationship. We use a bicentric design to validate the significance of predictors cross-contextually.

We demonstrate correlated change trajectories in tinnitus-related distress and depression that stay significant even when adjusting for additional covariates such as somatization and distress-related factors. This highlights the depressive symptomatology in individuals suffering from chronic tinnitus. We conclude that addressing depressive symptoms in patients with tinnitus enhances treatment response and vice versa, suggesting a bidirectional relationship between tinnitus-related distress and depression.

## Methods

### Datasets: patients and interventions

The first dataset (for analysis 1) included demographic, tinnitus, and mental health-related data from 500 patients with severe chronic subjective tinnitus who attended therapy at the ENT Department of JUH. Data were assessed at the beginning (pre) and end (post) of five consecutive treatment days between July 2013 and April 2017. Patients received an interdisciplinary day-care treatment[19].

The second dataset (for analysis 2) consisted of retrospective data from 1016 patients with severe chronic subjective tinnitus treated at the Tinnitus Center of the Charité Universitätsmedizin Berlin between January 2011 and October 2015. Treatment followed a multimodal 7-day concept[20]. Both treatment approaches focus on evidence-based interventions following the German S3 guideline for chronic tinnitus[7]. They involve interdisciplinary collaboration among specialists from psychology, audiology, otorhinolaryngology, and physical medicine and rehabilitation. Both approaches aim to achieve habituation and alleviate tinnitus-related distress through comprehensive otorhinolaryngological and psychological diagnostics, daily psychoeducational tinnitus counseling, CBT, mindfulness-based interventions, and physiotherapy, including relaxation exercises. The main differences between the two treatment regimens are the duration of treatment and the use of hearing aids. At JUH, all patients receive individually fitted hearing aids using a standardized $DSL_{child}$-algorithm-based setting[21]. At Charité Universitätsmedizin Berlin, the use of hearing aids is optional.

Inclusion criteria for treatment require the clinical diagnosis of chronic tinnitus with a minimum persistence of 3 months and clinically relevant tinnitus-related distress, as determined by a clinical psychologist. The psychological screening process at JUH requires a TQ score above 30, indicating moderate to severe levels of tinnitus-related distress, while patients at the Tinnitus Center of the Charité Universitätsmedizin Berlin are not required to meet a specific cutoff on the TQ scale, leading to a sample with slightly lower TQ scores pre-treatment compared to the JUH sample. This bicentric approach allows for testing whether the same predictors are relevant in tinnitus treatment, independent of the treatment setting and the questionnaires applied to measure change.

Patients with acute suicidality and severe psychiatric diagnoses, preventing participation in group therapy, are not included in treatment at both centers.

All patients were 18 years of age or older, proficient in the German language, and gave their informed written consent to participate in the study. All relevant ethical regulations were followed. The protocol was approved by the ethics committee of JUH (4366-03/15) and by the Charité Universitätsmedizin Berlin ethics committee (EA1/115/15) in accordance with the recommendations of ICH harmonized tripartite guideline for Good Clinical Practice, as well as the Declaration of Helsinki.

### Primary outcome measure

Tinnitus-related distress was assessed at the beginning of therapy (pre) and after treatment (post) with the German version of the Tinnitus Questionnaire (TQ)[22]. The TQ is frequently used as a standard psychometric instrument for assessing the severity of tinnitus annoyance and impairment. It shows evidence for high test-retest reliability and internal consistency[22]. Even though tinnitus severity questionnaires were originally not designed to be sensitive to treatment-related changes, they have been recommended to evaluate outcomes following interventions[23]. The overall sum score of all 52 items ranges between 0 and 84. A cutoff value differentiates between compensated (≤46 indicating low secondary symptoms and bothersome tinnitus under the experience of mental or physiological strain) and decompensated symptom levels (≥47 indicating ongoing tinnitus

annoyance and psychological distress from cognitive, emotional, and physiological impairment to severe psychological decompensation). Data acquired at post was used as the dependent variable in analysis 1.

The TQ and all accompanying questionnaires are summarized in Table 1. Differences in the applied questionnaires between Tinnitus Centers have been accommodated, referring to their conceptual similarities. If a construct was measured by multiple psychometric instruments, the individual questionnaire scores were transformed to a common scale and aggregated to create one compound score per construct.

## Statistics and reproducibility
**Analysis 1: Predictor selection.** The Elastic Net (ENet) is a regularization and variable selection method that enables model simplification without compromising the predictive accuracy of a model[24]. This machine learning approach is based on a linear regression model including two different penalty terms that shrink coefficients towards zero while only keeping important predictors in the model. The obtained parsimonious models offer improved interpretability and generalizability while lowering the risk of overfitting[25]. As a generalization of the RIDGE[26] or LASSO[27] regression, the ENet provides the advantage of increased prediction accuracy under collinearity due to a grouping effect[24]. Strongly correlated predictors are included in the model together without conceptual loss of information. For example, the LASSO regression would randomly select one variable from each group of highly correlated predictors.

To control the ideal amount of penalization, a suitable value for the tuning parameter λ has been selected by tenfold cross-validation. The mean square error (MSE) measures the model's prediction error by quantifying the average difference between actual and predicted values of the outcome variable. The value of λ associated with a minimal MSE was selected for the analysis. The remaining hyperparameter α was set to 0.5 to indicate a balanced ratio of both penalty terms.

Variables were z-standardized before the analyses. The ENet was applied to the data using *glmnet*[25] with the statistical analysis software R version 4.1.0.

## Analysis 2: Model generation from predictors using LCM.
LCMs allow for studying mean–level changes as well as individual differences in change trajectories within one model[28]. The bivariate LCM follows a multivariate extension that enables the inclusion of growth in two constructs simultaneously to examine whether change over time in one construct is related to the change trajectory in the other construct[17].

The latent growth factors (intercept and slope) are interpreted as individual differences in the level (initial status) and the rate of change in growth trajectories over time[17]. Here, the latent intercept represents the initial mean level of tinnitus-related distress at the beginning of therapy. The latent slope indicates the mean rate of change in tinnitus-related distress due to treatment (e.g., the negative slope of tinnitus distress with therapy success). To investigate a possible cause-and-effect association between depression severity and tinnitus-related distress, both constructs were operationalized as latent growth factors. Cross-domain relations are estimated as unidirectional regression parameters representing between-symptom treatment effects.

As factor loadings follow an increasing trend, higher scores on the slope factor suggest increases in tinnitus-related distress or depression severity, respectively. Thus, positive correlations/effects between intercept and slope would indicate that individuals with higher levels of tinnitus-related distress (or depression severity) at baseline would tend to have higher slope scores, denoting less treatment-related change in the intended direction. Negative correlations/ effects, on the other hand, would suggest the prediction of lower slope scores if higher severity levels had been present pre-treatment, suggesting more treatment-related improvement.

With increasing model complexity, additional covariates are analyzed. Time-invariant covariates vary across individuals but represent characteristics that stay constant over the time frame of interest (here: age and sex) and are used to predict the growth factors directly, whereas time-varying covariates (here: anxiety, perceived stress, and somatization) represent time-dependent influences and can be used as repeated exogenous predictors of variability across the outcome measures[17].

LCMs can be applied to data, including only two measurement waves[17]. Of note, growth trajectories are thus forced to be linear, regardless of the developmental trajectory's true shape. Due to the short time frame of interest, the linear representation of change trajectories was considered adequate. All models were estimated using the latent variable analysis package *lavaan*[29] implemented in R. Analyses were performed on all cases with complete data using a Maximum Likelihood (ML) estimator. Statistical significance was assumed if *p* < 0.05.

**Analysis plan.** First, an *unconditional* parallel process model was specified to assess correlational relationships between tinnitus-related distress and depression severity, as well as their change rates over time. In the *depression severity* model, predictive paths were added. The intercept of each latent construct was used to predict the within and across construct slope to determine whether baseline symptom severity is systematically associated with change and which prediction direction (initial tinnitus-related distress predicting the rate of change in depression severity or initial depression severity predicting the rate of change in tinnitus-related distress) might be more suitable. The *depression severity* model was then extended to assess the effect of time-invariant *demographic* covariates (sex and age differences) on baseline levels and change processes of tinnitus-related distress and depression severity. This model was extended further by adding time-varying covariates (anxiety, somatic symptoms, and stress), constituting a *mental health* model. This model reflects growth in the outcome variables controlling for occasion-specific effects of the time-varying covariates apart from the explanatory power of the predictors assessed at the same time point. To obtain a lagged effect, the covariates measured before treatment were used to predict tinnitus-related distress and depression severity after treatment to study whether baseline values of the predictor variables are associated with post-treatment measurements of the outcome variable. An extensive analysis of the first three models is provided in the Supplementary Results. Numerical results are provided in the Supplementary Tables 2–4.

**Inclusion and ethics statement.** All involved authors stem from three local regions (Jena, Berlin, and Münster). They fulfill the criteria for authorship and were included independent of their scientific or administrative status. The authors represent various scientific disciplines. The data were collected at two local university hospitals (JUH and Charité), most of which had patients originating from the respective cities or states. The global scientific community developed the administered treatments

## Results
**Analysis 1: Predictor selection.** All applied questionnaires are presented in Table 1. Clinical data of the JUH sample are summarized in Table 2. The sample included 243 female (48.6%) and 257 male (51.4%) participants between 22 and 80 years of age with an average of 55.2 years (SD = 11.5). On average, patients had been suffering from tinnitus for approximately eight years prior to treatment. Descriptive data show an average reduction of tinnitus-related distress by approximately 14.5 points (SD = 12.2) on the TQ scale. The number of patients whose TQ-score change resulted in compensated tinnitus after treatment can be obtained from Table 2. 280 patients (56%) exceeded the 12-point criterion change on the TQ scale required for clinical significance[30].

The ENet yielded the following baseline features in the best model for predicting tinnitus-related distress post-treatment with Root MSE = 11.35 ranked by decreasing predictive power: baseline tinnitus-related distress (TQ), age, female sex, somatoform symptoms (PHQ–15), depressive symptoms (PHQ–9) and perceived psychosocial stress (PHQ–10). The numerical coefficients are provided in Supplementary Table 1.

**Table 1 | Questionnaires used in each Tinnitus-Center (JUH and Charité Universitätsmedizin Berlin)**

| Construct | JUH Questionnaire (time of measurement) | Description | Charité Questionnaire (time of measurement) | Description |
|---|---|---|---|---|
| Tinnitus–related distress | TQ (pre, post) | Tinnitus Questionnaire[22], 52 items, 3-ary scale, sum score 0–84, cutoff: compensated ≤ 46, decompensated ≥ 47, sum score | TQ (pre, post) | Tinnitus Questionnaire, 52 items, 3-ary scale, sum score 0–84, cutoff: compensated ≤ 46, decompensated ≥ 47, sum score |
| Depression | PHQ-9 (pre) | Subscale depressive mood of the Patient Health Questionnaire[47] – long form (PHQ-D), 9 items, 4-ary scale, severity grades: none, mild medium, severe | ADS-L (pre, post) | Center of Epidemiologic Studies Depression Scale[48], 20 items, 4-ary scale, cutoff > 22 indicated the presence of a depressive disorder, sum score |
| | | | BSF (pre, post) | Subscales anxious depression (BSF_n) and an inversion of the elevated mood scale (BSF_l) of the Mood Adjective Check List[49], 5 items per scale, 5-level scale, sum scores |
| | | | PHQ-k (pre, post) | Subscale depressive mood of the PHQ short form, 9 Items, 4-level scale, sum score |
| | | | ISR_d (pre, post) | Subscale depressive syndrome of the ICD-10-Symptom Rating[50] (ISR_d), 4 items, 5-level scale, sum score |
| | | | Compound score: D | |
| Somatic symptoms | PHQ-15 (pre) | Subscale somatic symptoms of the PHQ-D, 15 Items, 4-ary scale, severity grades: none, mild medium, severe | BI (pre, post) | Berlin Complaint Inventory[51], 57 items, 5-level scale, sum score |
| | | | SF-8 (pre, post) | Physical component summary score of the Short Form-8 Health Survey[52], 8 items, 5/6-level scale, inverted sum score |
| | | | ISR_s (pre, post) | Subscale somatoform syndrome of the ICD-10-Symptom Rating (ISR_s), 3 items, 5-level scale, sum score |
| | | | Compound score: S | |
| Perceived stress | PHQ-10 (pre) | Subscale psychosocial stress of the PHQ-D, 10 Items, 4-ary scale, severity grades: none, mild medium, severe | PSQ (pre, post) | Perceived stress questionnaire[53], 20 items, 4-level scale, sum score |
| Anxiety | GAD-7 (pre) | Subscale anxiety of the PHQ-D, 7 Items, 4-ary scale, severity grades: none, mild medium, severe | ISR_a (pre, post) | Subscale anxious symptoms of the ICD-10-Symptom Rating (ISR_a), 4 items, 5-level scale, sum score |

Data were assessed at the beginning (pre) and end (post) of five (JUH) or seven (Charité) consecutive treatment days. If a construct was measured by multiple psychometric instruments, the questionnaire scores were combined into one scale by first re-inverting the sum scores with reversed polarity and transforming all variables to a scale between 0 and 100. Then a new compound score was created for each construct by averaging the sum scores across all scales within the same construct. All scores were centered and standardized.

**Table 2 | Distribution of patient assessments within the JUH sample, M ± SD (*N*)**

| | male 51.4% (257) | | female 48.6% (243) | | Total *N* = 500 (age = 55.2 ± 11.5 years) | |
|---|---|---|---|---|---|---|
| | pre | post | pre | post | pre | post |
| TQ | 48.89 ± 14.08 | 35.81 ± 15.38 | 49.21 ± 12.66 | 33.16 ± 14.81 | 49.05 ± 13.40 | 34.53 ± 15.15 |
| compensated | 43.6% (112) | 74.3% (191) | 39.5% (96) | 79.8% (194) | 41.6% (208) | 77.0% (385) |
| decompensated | 56.4% (145) | 25.7% (66) | 60.5% (147) | 20.2% (49) | 58.4% (292) | 23.0% (115) |
| PHQ-9 (Depressive mood) | 1.32 ± 0.93 | | 1.36 ± 0.82 | | 1.34 ± 0.88 | |
| mild | 38.1% (98) | | 46.5% (113) | | 42.2% (211) | |
| moderate | 30.0% (77) | | 31.3% (76) | | 30.6% (153) | |
| severe | 11.3% (29) | | 9.05% (22) | | 10.2% (51) | |
| PHQ-15 (Somatic symptoms) | 1.45 ± 0.96 | | 1.76 ± 0.87 | | 1.55 ± 0.94 | |
| mild | 36.6% (94) | | 35.4% (86) | | 36.0% (180) | |
| moderate | 29.2% (75) | | 35.8% (87) | | 32.4% (162) | |
| severe | 13.6% (35) | | 23.0% (56) | | 18.2% (91) | |
| PHQ-10 (Stress) | 0.67 ± 0.76 | | 0.88 ± 0.80 | | 0.77 ± 0.79 | |
| mild | 38.1% (98) | | 41.6% (101) | | 39.8% (199) | |
| moderate | 10.9% (28) | | 19.3% (47) | | 15.0% (75) | |
| severe | 2.3% (6) | | 2.5% (6) | | 2.4% (12) | |
| GAD-7 (Anxiety) | 1.09 ± 0.94 | | 1.29 ± 0.89 | | 1.19 ± 0.92 | |
| mild | 39.7% (102) | | 39.9% (97) | | 39.8% (199) | |
| moderate | 20.6% (53) | | 30.9% (75) | | 25.6% (128) | |
| severe | 9.3% (24) | | 9.0% (22) | | 9.2% (46) | |

Data were assessed at the beginning (pre) and end (post) of five consecutive treatment days. Percentages below the threshold for mild symptom severity have not been included in this table. Compensated (≤46; low secondary symptoms and bothersome tinnitus under the experience of mental or physiological strain), decompensated (≥47; ongoing tinnitus annoyance and psychological distress from cognitive, emotional, and physiological impairment to severe psychological decompensation).
*TQ* Tinnitus Questionnaire, *PHQ* Patient Health Questionnaire, *GAD-7* anxiety subscale of the PHQ, *M* arithmetic mean, *SD* standard deviation, *N* number of patients in each category.

**Analysis 2: Model generation from predictors.** Clinical data of all included 512 female (50.4%) and 504 (49.6%) male patients between 18 and 78 years of age with an average of 49.3 years (SD = 11.8) has been summarized in Table 3. On average, tinnitus-related distress was reduced by approximately 7.2 points (SD = 9.4) on the TQ scale after treatment. The number of patients with a TQ-score change leading from decompensated to compensated tinnitus after treatment can be obtained from Table 3. 29.82% of patients (*N* = 198) exceeded the 12-point criterion change on the TQ-scale required for clinical significance[30].

The parallel process *mental health* model, including tinnitus-related distress and depression severity as latent constructs as well as demographic and psychological covariates, provides adequate model fit with CFI = 0.993, TLI = 0.956, RMSEA = 0.079 (90%-confidence interval: 0.058 – 0.102, $p = 0.012$), and SRMR = 0.016, df = 6. Estimates and correlations can be obtained from Fig. 1.

Perceived stress, somatic symptoms, and anxiety are significantly associated with tinnitus-related distress at both time points ($p_{\text{TQ (pre)}\sim\text{PSQ (pre)}} = 0.000$, $p_{\text{TQ (post)}\sim\text{PSQ (post)}} = 0.000$, $p_{\text{TQ (pre)}\sim\text{S (pre)}} = 0.000$, $p_{\text{TQ(post)}\sim\text{S (pre)}} = 0.000$, $p_{\text{TQ (post)}\sim\text{S (post)}} = 0.000$, $p_{\text{TQ (pre)}\sim\text{ISR\_a (pre)}} < 0.05$, $p_{\text{TQ(post)}\sim\text{ISR (post)}} < 0.01$), except for a non–significant lagged effect in anxiety ($p_{\text{TQ (post)}\sim\text{ISR\_a (pre)}} = 0.505$) and a significant but inverse lagged effect in psychosocial stress ($p_{\text{TQ (post)}\sim\text{PSQ (pre)}} = 0.000$). These associations indicate that increased impairment in psychological covariates (besides the mentioned exceptions) is related to higher scores in tinnitus-related distress within and across both measurement waves. A comparison of the absolute estimate values suggests that somatic symptoms and perceived stress affect tinnitus-related distress to a higher degree than anxiety. Depression severity is also significantly associated with all psychological variables within one measurement wave ($p_{\text{D (pre)}\sim\text{PSQ (pre)}} = 0.000$, $p_{\text{D (post)}\sim\text{PSQ (post)}} = 0.000$, $p_{\text{D (pre)}\sim\text{S (pre)}} = 0.000$, $p_{\text{D(post)}\sim\text{S(post)}} = 0.000$, $p_{\text{D(pre)}\sim\text{ISR\_a (pre)}} = 0.000$, $p_{\text{D (pre)}\sim\text{ISR\_a(post)}} = 0.000$). The cross-lagged effects between depression severity and all

mental health variables are not significant ($p_{\text{D (post)}\sim\text{PSQ (pre)}} = 0.069$, $p_{\text{D (post)}\sim\text{S (pre)}} = 0.254$, $p_{\text{D(post)}\sim\text{ISR\_a (pre)}} = 0.362$).

Interindividual differences in both demographic covariates (age, sex) first suggest that older patients might start therapy with higher levels of tinnitus-related distress and improve less through treatment ($p_{\text{i\_t} \sim \text{age}} = 0.000$, $p_{\text{s\_t} \sim \text{age}} < 0.05$). The covariate biological sex is not associated with either tinnitus–related intercept ($p_{\text{i\_t} \sim \text{sex}} = 0.256$) or slope ($p_{\text{s\_t} \sim \text{sex}} = 0.255$) but with the rate of change in depression severity ($p_{\text{s\_D} \sim \text{sex}} < 0.01$), indicating that female patients might perceive more improvement in depressiveness over the course of therapy.

Compared to the *depression severity* and *demographic* model (for the preceding models, see Supplementary Results), correlative associations between tinnitus-related distress and depression severity decrease when controlling for the remaining mental health conditions (perceived stress, somatic symptoms, and anxiety) but stay significant ($p = 0.000$). In contrast to the preceding models, the between-symptom treatment effects are not significant when adding other psychological variables to the model (initial depressiveness predicting change in tinnitus-related distress due to treatment $p = 0.371$; initial tinnitus-related distress predicting the rate of change in depression severity $p = 0.706$).

Both findings suggest a common factor between tinnitus perception, depressiveness, and the psychological covariates that might explain part of their associations.

## Discussion

In analysis 1, a regularization procedure was conducted on the JUH dataset to select a relevant subset of variables for the prediction of therapy success and to investigate the relationship between tinnitus-related distress and depression severity. The obtained predictors *tinnitus-related distress, somatic symptoms, depressiveness, psychosocial stress, anxiety, age,* and *sex* were selected for analysis 2. LCMs with increasing complexity were estimated to investigate the relationship between tinnitus-related distress and

**Table 3 | Distribution of patient assessments within Charité Universitätsmedizin Berlin sample; M±SD (N)**

|  | male 49.6% (504) | | female 50.4% (512) | | Total N = 1016 (age = 49.3 ± 11.8) | |
|---|---|---|---|---|---|---|
|  | pre | post | pre | post | pre | post |
| TQ | 37.90 ± 17.54 | 30.98 ± 17.67 | 39.87 ± 15.94 | 32.37 ± 16.13 | 38.89 ± 16.77 | 31.68 ± 16.91 |
| compensated | 67.9% (342) | 81.2% (409) | 66.8% (342) | 79.7% (408) | 67.3% (684) | 80.4% (817) |
| decompensated | 32.1% (162) | 18.8% (95) | 33.2% (170) | 20.3% (104) | 32.7% (332) | 19.6% (199) |
| ADSL | 16.62 ± 11.46 | 12.47 ± 10.21 | 19.50 ± 11.17 | 13.87 ± 10.98 | 18.01 ± 11.40 | 13.18 ± 10.62 |
| Depr. disorder | 29.2% (147) | 19.3% (97) | 39.6% (203) | 22.1% (113) | 34.4% (350) | 20.7% (210) |
| ISR_s | 0.62 ± 0.81 | 0.58 ± 0.78 | 0.58 ± 0.76 | 0.58 ± 0.77 | 0.60 ± 0.78 | 0.58 ± 0.77 |
| mild | 8.7% (44) | 8.1% (41) | 8.6% (44) | 11.1% (57) | 8.7% (88) | 9.6% (98) |
| medium | 17.7% (89) | 17.7% (89) | 18.4% (94) | 16.2% (83) | 18.0% (183) | 16.9% (172) |
| severe | 3,6% (18) | 3.0% (15) | 2.1% (11) | 3.1% (16) | 2.9% (29) | 3.1% (31) |
| ISR_a | 0.81 ± 0.89 | 0.77 ± 0.87 | 1.07 ± 0.90 | 0.99 ± 0.92 | 0.94 ± 0.90 | 0.88 ± 0.90 |
| mild | 21.0% (106) | 22.4% (113) | 33.0% (169) | 28.5% (146) | 27.1% (275) | 25.5% (259) |
| medium | 8.9% (45) | 10.9% (55) | 14.3% (73) | 13.3% (68) | 11.6% (118) | 12.1% (123) |
| severe | 4.2% (21) | 2.6% (13) | 4.1% (21) | 4.9% (25) | 4.1% (42) | 3.7% (38) |
| PSQ | 0.44 ± 0.18 | 0.41 ± 0.19 | 0.49 ± 0.17 | 0.45 ± 0.18 | 0.46 ± 0.18 | 0.43 ± 0.19 |

The ADSL sum score and subscale scores of ISR_s and ISR_a have been reported as exemplary indicators for the prevalence of depression, somatic symptoms, and anxiety in the present sample. Data were assessed at the beginning (pre) and end (post) of seven consecutive treatment days. Percentages with minor or no symptomatic (ISR) have not been displayed in the table. Compensated (≤ 46; low secondary symptoms and bothersome tinnitus under the experience of mental or physiological strain), decompensated (≥47; ongoing tinnitus annoyance and psychological distress from cognitive, emotional, and physiological impairment to severe psychological decompensation).

TQ Tinnitus Questionnaire, ADSL Center of Epidemiologic Studies Depression Scale, ISR_s subscale somatoform syndrome of the ICD-10-Symptom Rating, ISR_a subscale anxious symptoms of the ICD-10-Symptom Rating, PSQ Perceived stress questionnaire. M arithmetic mean, SD standard deviation, N number of patients in each category.

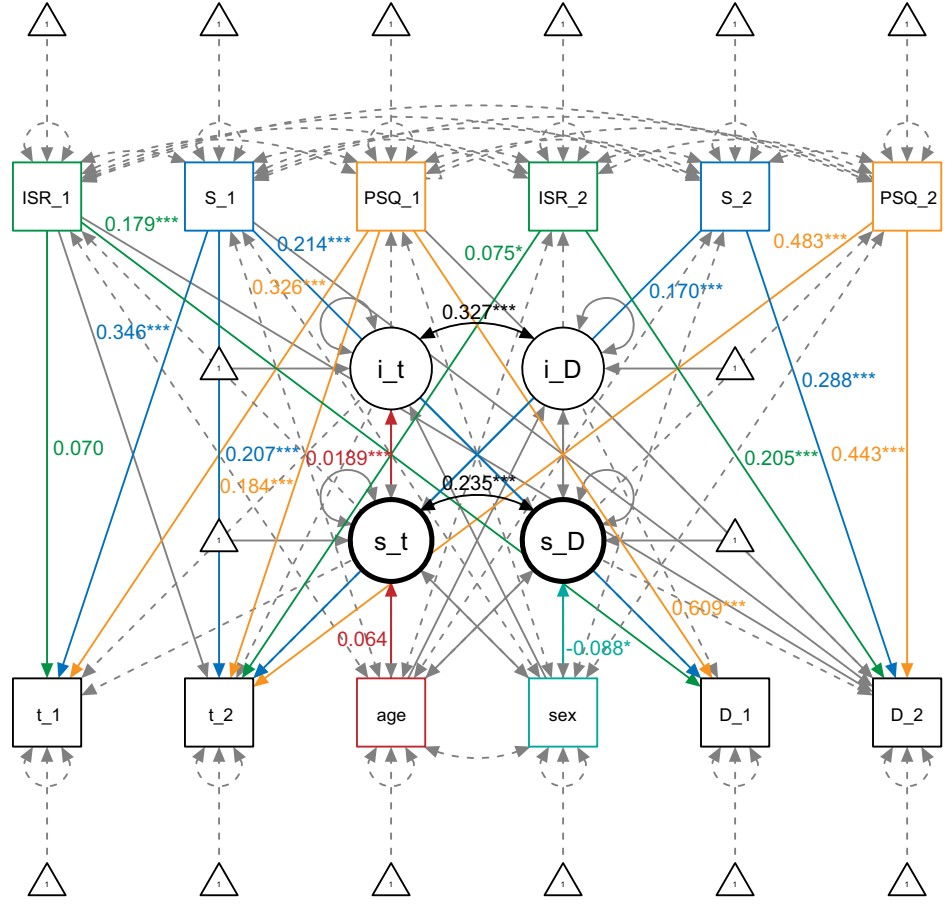

**Fig. 1 | Mental health LCM.** Circles = latent factors, squares = manifest measures, curved double-headed arrows = correlations/ covariations, straight one-headed arrows = predictive paths, i_t initial tinnitus distress, i_D initial depression severity, s_t rate of change in tinnitus distress, s_D rate of change in depression severity, S_1 somatic symptoms compound score (including SF8, BI_c, ISR_s) pre-treatment, S_2 somatic symptoms compound score post-treatment, PSQ_1 PSQ sum score pre-treatment, PSQ_2 PSQ sum score post-treatment, ISR_1 ISR a sum score pre-treatment, ISR_2 ISR_a sum score post-treatment, t_1 TQ score pre-treatment, t_2 TQ score post-treatment, D_1 depression severity compound score (including ADSL, BSF_n, BSF_l, PHQK, ISR_d) pre-treatment, D_2 Depression severity compound score post-treatment, significance scores: 0 '***' 0.001 '**' 0.01 '*' 0.05 '.' 0.1 ' '.

**Fig. 2 | Depression severity LCM.** Circles = latent factors, squares = manifest measures, curved double-headed arrows = correlations/ covariations, straight one-headed arrows = predictive paths, i_t initial tinnitus distress, i_D initial depression severity, s_t rate of change in tinnitus distress, s_D rate of change in depression severity, t_1 TQ score pre-treatment, t_2 TQ score post-treatment, D_1 depression severity compound score (including ADSL, BSF_n, BSF_l, PHQK, ISR_d) pre-treatment, D_2 Depression severity compound score post-treatment.

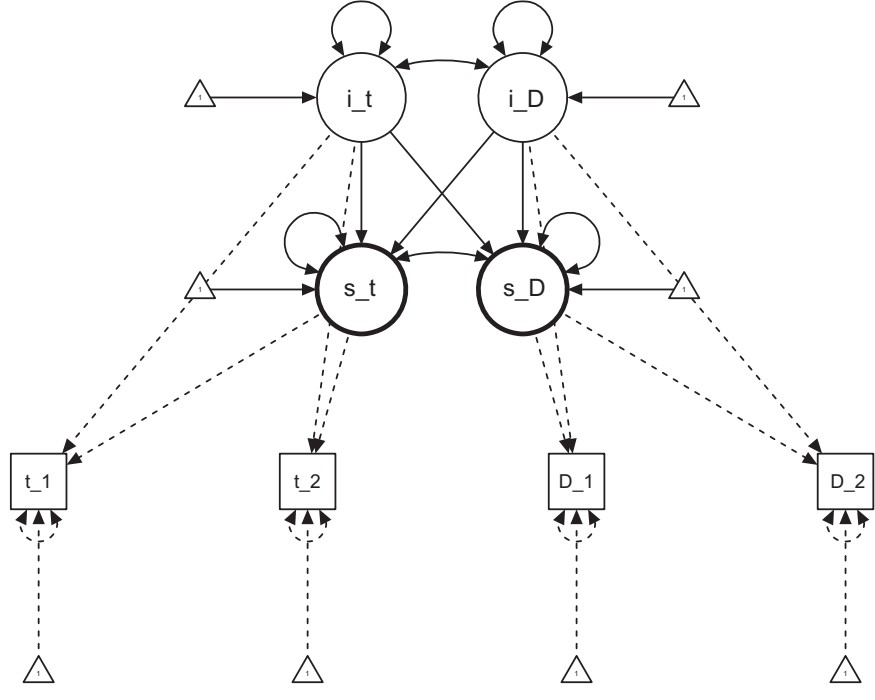

depression severity, considering the effect of the preselected mental health and sociodemographic variables. The main finding consists of a bidirectional change process in tinnitus-related distress and depression severity, as demonstrated by the association of latent slopes. This indicates that a targeted reduction in tinnitus-related distress is linked to a decrease in depression severity and vice versa.

## Analysis 1: Predictor selection

The positive regression coefficient of the TQ score before treatment suggests that patients with a high initial symptom severity also reported higher tinnitus annoyance after treatment. Also, all mental health-related predictors show a positive relationship with the outcome variable, indicating increased tinnitus-related distress after treatment when psychological strain due to somatization, psychosocial stress, and depressiveness had been present at baseline.

Although anxiety has been discussed as one of the most commonly related comorbidities of tinnitus[6], here GAD-7 has not been identified as a relevant predictor. This result supports previous findings that depressive symptoms entirely mediate the predictive effect of anxiety on tinnitus handicap[6]. However, to revisit the influence of anxiety on the association of tinnitus-related distress and depression severity, it has been included as a covariate in analysis 2.

## Analysis 2: Model generation from predictors

The bidirectional link between both latent variables is illustrated in Fig. 2. Across all investigated bivariate LCMs, the latent intercepts and slopes of tinnitus-related distress and depression severity were correlated significantly. From a psychopharmacological point of view, the beneficial effect of antidepressants on tinnitus-related distress reflects the correlated change trajectories demonstrated here[31]. The close association of tinnitus-related distress and depression severity at baseline is compatible with earlier findings on an elevated prevalence of depressive symptoms in patients with tinnitus[32].

In the *depression severity* and *demographic* model, the cross-lagged treatment effect of baseline depressiveness on tinnitus-related distress post-treatment further emphasizes the role of depression in maintaining tinnitus-related distress and mitigating treatment success. Tinnitus-related distress might also contribute to developing and perpetuating depressive symptoms. However, when controlling for other psychological variables within the

*mental health* model, the predictive association between baseline symptom severity and treatment change is not significant anymore. The correlative association between latent intercepts and slopes is also lowered, although still statistically significant, suggesting that the association between tinnitus-related distress and depression severity can partly be explained by somatization and distress-related components. Ooms et al.[16], for instance, argue that tinnitus-related distress and depression are not substantially related but connected by a methodological artifact that can be explained by somatic symptoms. From a clinical perspective, Langguth et al.[13] describe a different symptomatic spectrum of depressive tendencies in patients with tinnitus compared to a major depressive episode. While symptoms such as fatigue, irritability, insomnia, exhaustion, and concentration difficulties are present within the perception of patients with tinnitus, other core symptoms imperative for the diagnosis of depression (depressed mood or loss of interest), are often absent. While depression seems to equally affect patients with severe tinnitus-related distress and participants with a diagnosed depressive disorder, somatization is reported as a factor only related to patients with tinnitus[33]. The importance of psychosomatic symptoms in tinnitus might reflect a possible disposition of tinnitus patients that includes increased maladaptive associations with body, health, or sensory perceptions leading to maintained tinnitus awareness and increased perceived intrusiveness of the tinnitus sound[32]. The close relationship between tinnitus severity and stress, also demonstrated previously[32,34], enhances the relevance of state-dependent stress perception for the moment-to-moment prediction of tinnitus severity as a target in intervention concepts[35].

Augmented tinnitus-related distress at baseline and increased maintenance of tinnitus-related distress with increasing age have previously been noted in several studies[36–38] (for exceptions, see ref.[39–41]). Normatively declined compensatory brain plasticity in older age might lead to maladaptive coping and the reduced ability to suppress the permanent perception of tinnitus-related symptoms[42]. Also, increased age-dependent auditory perceptual difficulties might affect tinnitus perception and severity while influencing patients' emotional well-being[41]. Hearing impairment is assessed within the TQ, and older patients might score higher on the corresponding items due to symptoms of presbyacusis[12].

In the *mental health* model, sex differences were not associated with baseline tinnitus-related distress or the rate of change due to treatment, which aligns with earlier findings[40,41]. However, the significant association between sex and the change rate in depressiveness might insinuate gender

differences in emotional regulation that might lead to differential responses to the perception of tinnitus[11,12,43].

Although anxiety has long been proposed as a cause and amplifier of tinnitus severity[32], the association between tinnitus-related distress and anxiety was comparatively low in the *mental health* model. According to a recent meta-analysis, mixed evidence on the presence of anxiety in tinnitus patients may indicate that other factors modulate this association[32]. As the relationship between tinnitus impairment and anxiety was demonstrated to be fully mediated by somatic and cognitive aspects of depression[6], the weak association between tinnitus-related distress and anxiety is explained through the remaining associations between the latent factors of tinnitus-related distress and depression severity.

### Limitations
While the ENet is recommended for its performance under multicollinearity, high correlations among predictors, as seen here among psychological features, can still lead to instability in coefficient estimates and inflate standard errors, which may affect the model's performance[44]. Despite the large sample size and the application of an effective method for feature selection, the high Root MSE, therefore, emphasizes the difficulty of predicting tinnitus-related distress[11]. Due to penalization, all regression coefficients are biased towards zero and cannot be interpreted directly. Even though this method does not lead to reliably and directly applicable prediction models, it can be considered valuable to effectively separate relevant predictors from irrelevant features in a data-driven approach informing further analyses.

As the precision and reliability in growth measurements increase with the number of observations per individual[17], the quality of research findings would have been enhanced if more than two measurement waves had been available for LCM analysis. Because of the limited number of observations, the accuracy of parameter estimates might be compromised. Future research might examine a larger number of measurement waves, including data from preliminary screenings and follow-up consultations, to study antecedents and consequences of change due to therapy as well as predictors of a maintained or continued reduction in tinnitus-related distress post-treatment. Few similarities in the applied measurements at both sites further stress the importance of harmonized assessment strategies in future tinnitus research.

### Conclusions
Here, we establish the bidirectional association of tinnitus-related distress and depression severity across treatment, highlighting the beneficial effect of addressing depressive symptoms in patients with tinnitus and vice versa. If patients with a primary diagnosis of depression also report tinnitus, our findings stress the importance of addressing the experienced tinnitus-related distress as part of their treatment. A combined treatment of both conditions is recommended under a patient-centered and individualized perspective on systems medicine[45]. We also show that tinnitus-related distress and its change due to treatment can sufficiently be predicted by a subset of psychological variables, including symptoms of depression, somatization, and perceived stress. While the relationship between tinnitus-related distress and depressiveness partially arises due to somatic aspects of depression and distress-related psychosomatic symptoms, this work reveals a core depressive common symptomatology that remains even after the addition of other psychological covariates and is relevant for the initial severity and the maintenance of tinnitus-related distress post-treatment. Somatization– and distress–related components within the depressive symptom spectrum are therefore especially relevant in predicting therapy success and should be considered in effective treatment strategies. The relevance of all included mental health-related variables across treatment settings contributes to the conception of tinnitus as a multifactorial symptom requiring multimodal treatment approaches.

### Reporting summary
Further information on research design is available in the Nature Portfolio Reporting Summary linked to this article.

### Data availability
The data are not publicly available due to them containing information that could compromise research participant privacy/consent. Anonymized data are available from the corresponding author CD upon reasonable request. Data are located in controlled access data storage at the University Hospital Jena. The source data for Fig. 1 can be found in Supplementary Data 1.

### Code availability
Analytic R code for both analyses is available here: https://doi.org/10.5281/zenodo.14045072[46].

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

## Acknowledgements

The authors thank Prof. Franz Neyer and Dr. Sebastian Pusch for their valuable comments and suggestions on data analysis. The authors disclose support for the research and publication of this work from the German Research Foundation (DO 711/10-1, 10-3; JU 445/10-3).

## Author Contributions

C.L., C.D., B.M. and M.J. designed the study. C.L. performed the statistical analyses, interpreted the results, and wrote the manuscript under the supervision of C.D. B.M., P.B., and O.G.L. collected the data. All authors edited the manuscript and reviewed it critically for important intellectual content.

## Funding

## Competing interests

The authors declare no competing interests.

## Additional information

**Supplementary information** The online version contains Supplementary Material available at https://doi.org/10.1038/s43856-024-00678-6.

