## [Transparent Peer Review file · Communications Medicine]

A retrospective two-center cohort study of the bidirectional relationship between depression and tinnitus-related distress

Corresponding Author: Professor Christian Dobel

Version 0:

Reviewer comments:

Reviewer #1

(Remarks to the Author)

The authors conducted a retrospective cohort study to examine the relationship between tinnitus-related distress and depression in two independent cohorts with tinnitus receiving therapy in Germany. They used a regression-based machine learning method to select predictors of treatment success using data from Cohort A (Jena University Hospital, n=500) which were applied to the data from Cohort B (Charite Universitätsmedizin Berlin, n=516) in a latent growth curve model. I am reviewing an updated version following revision to address comments from another journal. The authors have partially addressed my prior comments.

ABSTRACT

1. Regarding the term "daycare unit," please at least put "outpatient clinic" in parentheses to avoid confusion.
2. As in my prior review of this paper, please add to the abstract:
 - The number of patients in the datasets and the selection criteria—did all patients have severe chronic subjective tinnitus?
 - The years of data collection
 - The global region of data collection
 - That the therapy received was one of two forms of habituation therapy for severe chronic subjective tinnitus
 - The duration of therapy in each of the two cohorts
 - How distress and depression were assessed (what surveys)
 - The numerical results of 'weaker' associations
 - How sex was specifically related to the rate of change in depression

The current abstract is 161 words, and the limit is 250, so there should be enough words to briefly describe the study details.

INTRODUCTION

1. It needs to be clear that the form of tinnitus being studied is subjective chronic tinnitus. Otherwise, general statements like in Line 27-28 (about curative treatment) are incorrect as other forms may be curable. If the form is more specific than that (severely bothersome) that should be clarified.

2. Line 18-20: The authors should recognize the impact of geographical location on tinnitus incidence and prevalence, as recently summarized by Jarach et al. JAMA Neurol. 2022.

3. Line 28: The guideline needs to be cited. This needs to be clarified that it is a German guideline.
<https://doi.org/10.1007/s00106-022-01207-4>

4. Lines 50-52: While methodological details of the therapy received are fine to keep in the Methods section, it is important that the general type of treatment be mentioned in the Introduction, particularly given this article's placement of the Methods at the end and the stated aim of informing treatment strategies. Please add that the centers' treatment included two forms of tinnitus habituation therapy (or another descriptor). Similarly, add that these patients had clinically confirmed tinnitus-related distress.

RESULTS/DISCUSSION

1. The article lacks a table including the demographic details of the two patient populations. Age and sex are summarized for the cohort in various text locations but the clarity of the article would benefit from including the baseline characteristics of the

two populations in a table, or a dedicated first paragraph in the Results describing the two populations.

2. The use of "t1" and "t2" would be better labeled in the tables with the exact timepoints assessed ("start of therapy" / "end of therapy" in the headings)

3. I maintain that the integration of the Results and Discussion sections is confusing. If the process of predictor selection needs to be described first, then the Methods should be placed before the Results for clarity. This journal accommodates that. A separate Discussion section would also allow the authors to better highlight the novelty of the study and break up large blocks of text with impede readability. The limitations should be placed in this section together, for all analyses.

4.

Reviewer #3

(Remarks to the Author)

In addition to a reading of the revised MS, this reviewer considered the comments from the original reviewer #2. One area in particular remains a concern, as the reviewer comment starting, "The interpretation of associations..." was not addressed by the authors. My interpretation of the comment would center, at least in part, on the authors providing a few examples from their analysis that could point to specific intervention considerations that could enhance the reader's understanding of the results, and perhaps increase the putative benefit for reader-clinicians. For example, the two sites employed different intervention doses; were there any site-specific elements of intervention that might have contributed to the difference observed regarding TQ score changes (ie., the JUH scores improved by about 2x those of Charite)? This reviewer can only speculate regarding the original reviewer's thinking; nevertheless, the suggestion above may merit consideration.

Main:

Consider rewording a few passages.

Line 27: consider wording change to, "...psychotherapeutic interventions, and sound-based interventions aimed at alleviating tinnitus-related distress as opposed to eradicating the tinnitus sound."

Lines 31-32: consider wording change to, "...and related comorbidity hinders attempts at predicting tinnitus-related distress and treatment response [8]. Additionally, the application..."

Lines 36-37: consider wording change to, "...psychological features may be more strongly associated with tinnitus severity than measures purporting to characterize the tinnitus sound."

Line 66: Would "interdisciplinary" be more appropriate than "transcontextual"?

Results/Discussion:

This reviewer agrees with reviewer 1's comments from the original MS in that the shifting from results to discussion within each section is a bit confusing. I understand the authors' rationale, however the current structure impedes somewhat the flow of the article.

Line 73 (and elsewhere): The authors need to report in either the methods or results sections the 12-point criterion change on the TQ required for clinical significance. Their reported 14.5 point change exceeds the criterion value, however it would be of greater value to the reader if the percentage of participants whose scores exceeded the criterion value were also reported. For example, an average of 14.5 change COULD mean that everyone in the study experienced significant improvement, even as some participants remained in the decompensated condition. It might also be true (particularly at Charite) that participants switched from decompensated to compensated even if their score change did not exceed the criterion. Tables 2&3 provide the reader with the number of participants whose score changes resulted in "compensated" tinnitus, but the reader does not know how many participants exceeded the criterion value for improvement. Perhaps the authors could comment on the number of participants whose classification changed from decompensated to compensated, but whose TQ score change did not exceed 12 points. At the very least, the authors could report % of participants whose TQ scores showed significant improvement.

Conclusion:

Line 250: this reviewer prefers "symptom" to "syndrome" when labeling tinnitus, as the latter connotes phenotypic presentations that may be clinically validated in ways that tinnitus cannot.

Reviewer #4

(Remarks to the Author)

The authors investigated the relationship between tinnitus-related distress and depression and their bi-directional relationship. The data presented was collected from two sites, with data from one site used for predictor selection. The predictors were then used with data from the second site to investigate changes in tinnitus-related distress and depression at two timepoints during therapy. Findings show a bidirectional association between tinnitus-related distress and depression severity, across treatment.

The existence of distress and depression in many patients with tinnitus is known, and the novelty of this study appears to be establishing the bidirectional relationship. I believe the contribution should be clarified to better acknowledge previous

knowledge in the field.

As the study did not aim to assess efficacy of treatments as a main outcome, the last line of the abstract could be edited to clarify that "enhanced therapy success in depression when tinnitus-related distress is addressed and vice versa", is proposed for future treatment strategies (as in this study there was no comparison of different treatment strategies).

When selecting predictors (analysis 1), the authors include anxiety and exclude lateralization, contrary to what the predictor selection shows. This decision appears somewhat arbitrary, and it would seem more reasonable, in terms of the analysis methods, to report results without including/ removing the mentioned factors.

Line 76: The authors mention ENet resulted in features in chronological order. It is not clear what they mean by chronological in this context.

In the description for Table 1, they highlight that scores were transformed to a common scale. How is the transformation done?

It is unclear where some of the results are shown. For example, line 91 mentions the positive regression coefficient of the TQ score at t1 but it is unclear where the coefficients are reported.

Version 1:

Reviewer comments:

Reviewer #1

(Remarks to the Author)

The authors have largely responded to my prior comments, which has enhanced the clarity of the manuscript. For the newly organized Discussion section, it would benefit from a more traditional article format such as a starting paragraph which summarized the main findings before delving into the detailed interpretation of the two analyses. Breaking up long paragraphs can also help with the ease of reading of that section. My lingering criticism is the lack of numerical/precise reporting of the results in the Abstract, which instead uses the majority of the word count to make general statements. If this is typical for the journal's format, that's fine. Publication is recommended.

Reviewer #3

(Remarks to the Author)

The authors are to be commended for addressing reviewer comments, and in particular, comments from a newly-recruited reviewer. It is expected that this MS will be of interest to the readership.

Reviewer #4

(Remarks to the Author)

Thank you to the authors. My comments have been addressed.

Universitätsklinikum Jena • HNO/Phoniatrie • 07747 Jena

To the editor of Communications Medicine

Univ.-Prof. Dr. rer.soc. Christian Dobel
HNO-Klinik, Am Klinikum 1, D-07747
Jena Telefon: 03641 - 9329410 Telefax:
03641 - 9329306
E-Mail: christian.dobel@med.uni-jena.de
Web: <http://www.uniklinikum-jena.de/hno>

Jena, den 23. April 2024

Dear Dr. Cunha,

We wish to submit our research article, "Bidirectional relationship between depression and tinnitus-related distress: a retrospective two-center cohort study" for consideration by Communications Medicine. We previously submitted this article to Nature Mental Health (NATMH-23-0591), and the chief editor strongly recommended that we transfer our manuscript to Communications Medicine as a suitable venue for this work and for your editorial evaluation.

Tinnitus is a prevalent symptom representing a relevant public health concern associated with severe mental-health-related comorbidities, such as depression. While the association between tinnitus-related distress and depression is well-established, the underlying mechanisms are still not fully understood.

Our research investigates the dynamic co-development of tinnitus and depression using large datasets from two tinnitus-specific, multidisciplinary treatment centers. We employ advanced statistical methods to examine and reveal underlying common features that translate research findings to clinical practice by informing treatment strategies for both conditions.

Our results strongly suggest a bidirectional association between tinnitus-related distress and depression severity, indicating depressive mood as a key factor in tinnitus treatment and vice versa. While current guidelines recommend cognitive-behavioral therapy for tinnitus treatment, the crucial role of depression is not sufficiently emphasized. Additionally, tinnitus is not routinely addressed in the treatment of depression, even though patients suffering from depression frequently report tinnitus.

According to our research findings, a combined treatment of both conditions is crucial for therapy success under a patient-centered and individualized perspective on systems medicine, highlighting the importance of interdisciplinary treatment approaches and discourse.

The proposed manuscript is not under consideration or in press elsewhere. Thank you for your interest in the topic and our manuscript. Please find enclosed the feedback provided by the previous reviewers, along with our responses to address their comments.

Sincerely,

Cosima Lukas and Christian Dobel, on behalf of the co-authors

Poliklinik/Phoniatrie
Mo-Fr: 08:00 - 11:00 Uhr
Tel.: 03641 - 9329393
Tel.: 03641 - 9329394

Privatsprechstunde
Di: 11:00 - 14:30 Uhr
Tel.: 03641 - 9329301
Fax: 03641 - 9329302

Fachsprechstunden
Tel.: 03641 - 9329393
Tel.: 03641 - 9329394

Tumorsprechstunde
Mo: 08:00 - 13:00 Uhr

Speicheldrüsenerkrankungen
Di: 08:00 - 14:30 Uhr

Fazialis-Nerv-Zentrum
Elektrophysiologie
Di: 10:00 - 15:30 Uhr

Schlafbezogene Atemstörungen
Di: 12:00 - 15:00 Uhr

Implantierbare Hörsysteme
Mi: 10:00 - 13:00 Uhr

Funktionell-ästhetische
Chirurgie
Do: 12:00 - 15:00 Uhr

Ohrsprechstunde
Fr: 10:00 - 12:00 Uhr

Allergiesprechstunde
Fr: 13:00 - 15:00 Uhr

Riech-/Schmeckstörung
nach Vereinbarung

Stationen
A130
Tel.: 03641 - 9326130
A230
Tel.: 03641 - 9326230
E230 (Kinder)
Tel.: 03641 - 9328230

We are grateful to the reviewers for their time and comments. We have taken their valuable feedback into consideration and addressed it below in bold font. The manuscript has greatly benefited from the reviewers' constructive suggestions.

Reviewer#1 had the impression that we investigated novel therapeutic approaches, so they asked for details of treatment. Both of our centers exist for over 10 years, and the details of their therapeutic approaches have been published several times. In fact, as Reviewer#2 pointed out, we were interested in the dynamic interaction of mental-health-related factors and tinnitus-related distress.

Reviewers' comments will be in bold font, whereas our responses will be in light font.

Reviewer #1:

to the Editor OVERALL

The authors conducted a retrospective cohort study to examine the relationship between tinnitus-related distress and depression in two independent cohorts with tinnitus receiving therapy in Germany. They used a regression-based machine learning method to select predictors of treatment success using data from Cohort A (Jena University Hospital, n=500) which were applied to the data from Cohort B (Charite Universitätsmedizin Berlin, n=516) in a latent growth curve model. There is a long-standing known relationship between tinnitus and depression/distress in the literature, so the novelty of the analysis is tempered by lack of clarity as to the unique contribution of this study. It appears that the main finding would be most related to the efficacy of the treatment regimen used by the two cohorts, although there is an overall lack of explanation or detail about those treatments.

By utilizing a bicentric approach, large datasets, and current statistical methods, this work has demonstrated the bi-directionality of the interaction between tinnitus-related distress and depression severity during treatment, which has been hypothesized but never thoroughly analyzed. Furthermore, this approach has allowed us to show that the same psychological factors are relevant for predicting change trajectories in tinnitus treatment, independent of the treatment setting and questionnaires used.

- Could use editing for English grammar, wording, and improved clarity.

We made a sincere effort to ensure that the text was well-written prior to submission. We appreciate constructive criticism and are open to reviewing the text upon the editor's recommendation.

- Poorly organized and needs to make use of subheadings under main headings.

We have carefully reviewed each subheading and found the length of the sections appropriate. Following one of the comments below, we restructured the subheadings in the methods section. We are open to adjusting the overall structure upon the editor's request.

- There are a number of undefined abbreviations at first use, and a number of previously abbreviated words that did not use the abbreviation. Please review and correct.

This has been changed as suggested.

- Many general or unclear vs. specific statements with appropriate citations and details (i.e., years, magnitude, numerical findings).

To improve readability, all numerical findings are included in the tables.

- Patient-first language is generally preferred (patients with tinnitus and not "tinnitus patients"). Sex and gender should not be used interchangeably.

“Tinnitus patients” has been replaced by “patients with tinnitus” throughout the entire text. The authors strongly agree that sex and gender are distinct constructs and should be considered continuous and dimensional. However, when referring to the existing literature, we had to adopt the phrasing used there.

- It needs to be made clear that the study populations had received clinical diagnosis of tinnitus at the time of study initiation, and that this is a population in Germany treated with specific regimens.

In response to Reviewer #2's comments, a detailed explanation of treatment regimens has been added. The text clearly outlines the location of the treatment centers and the inclusion criteria.

- The therapy received by the cohorts is not described and is instead only cited in the Methods. Please describe the therapy received by both cohorts of patients upfront in the Introduction.

The primary objective of this research is to examine how tinnitus-related distress and depression interact during therapy rather than assessing the efficacy of the treatment regimens. To maintain a concise introduction to the aim of this work, we would prefer to keep the detailed description of treatments in the methods section.

ABSTRACT

- The word “daycare units” is confusing; suggest using a more specific term like “outpatient otologic clinic”

In previous publications, the JUH Tinnitus Center has been specifically introduced as a “daycare unit.” In our opinion, this term best reflects the treatment concepts of both centers.

- State the number of patients in the datasets and the selection criteria—did all patients have subjective chronic tinnitus?

- State the years of data collection

- State the global region of data collection

- State the duration of therapy

- State how distress and depression were assessed (what surveys)

We acknowledge the importance of this information, but we are constrained by the word count limitations for abstracts. All relevant details are included in the methods section and can be added to the abstract upon the editor's recommendation.

- Define all abbreviations used

All abbreviations are defined. The model fit indices are commonly used and, in our view, do not require additional clarification.

- State the numerical results of ‘weaker’ associations

- Be more specific in stating how sex was related to the rate of change in depression

We aim to keep the abstract focused on the study's primary objective, given the limited word count. Therefore, both secondary findings are discussed in detail in the main text. If necessary, we could consider removing the secondary findings from the abstract to avoid any ambiguity.

INTRODUCTION

- Line 18: The use of “recent” is relative. Please state the date of any statistical data from the literature.

This has been changed as suggested.

- Line 19: The prevalence rates of tinnitus and severe tinnitus should not come from a single publication but should be synthesized, preferably using global data. As a subjective condition that is primarily assessed with self-report surveys, the rates can differ widely depending on the definition of tinnitus and population surveyed. This needs to be recognized and a broader view taken.

This has been resolved by recognizing the variability in prevalence estimates.

- Line 22: The proportion of patients with tinnitus-related distress should be stated from the literature, and the relationship between severe tinnitus and distress should be discussed

This has been changed as suggested.

- Line 24: Not all treatments for tinnitus are related to distress specifically, such as masking or hearing aids, and should be mentioned

This information is now included in the text. According to the current S3 guideline for chronic tinnitus, hearing aids are recommended due to their ability to relieve tinnitus-related distress and improve habituation (<https://doi.org/10.1007/s00106-022-01207-4>).

- Line 25: The names of clinical treatment guidelines need to be stated in the manuscript when referred to

This has been changed as suggested.

- Line 32: "Since" should be "Because

This has been changed as suggested.

- Line 34: What is the definition of treatment success? Which treatments?

This has been rephrased.

- Line 37: Year of systematic review data? When referring to populations or other region-specific prior research, state the region it was assessed in (global, European, etc.)

This has been changed as suggested.

- Line 38-39: Tinnitus and depression or tinnitus-related distress and depression? This is unclear

This has been clarified.

- Line 48: Duration of therapy? What therapy specifically?

- Line 49: An outpatient otologic clinic? How many patients?

- Line 53: How many patients were in the external cohort? Were they age and sex-matched? The other tinnitus center should be named specifically.

To maintain a concise introduction, we prefer specifying all treatment—and patient-related information in the methods section. In lines 59 and 60, the names of both Tinnitus Centers are mentioned.

RESULTS

- There are no tables describing the baseline characteristics of the cohorts, please add these. Also include the proportions with severe tinnitus, tinnitus-related distress at baseline, and depression at baseline

It appears there is a misunderstanding. This is reported in the results section. The corresponding proportions are already covered in Tables 2 and 3. Table 2 has been renamed for clarification.

- Table 1: please correct misspellings; please explain what compound score refers to (in the caption). When are “t1” and “t2” timepoints? These need to be defined in the caption.

This information has been added to the caption. The typo has been corrected.

- Table 2: “Compensated” and “decompensated” need to be explained in the caption. Tinnitus localization information in the caption should be moved to patient baseline characteristics tables. “T1” and “t2” timepoints need to be defined in the caption. Why are data missing at T2 for the non-tinnitus tests?

The terms “compensated” and “decompensated”, as well as the time points, are now defined throughout all tables, in addition to the explanation in the methods section.

The PHQ was only assessed once in the JUH sample. Therefore, we have used the JUH dataset for predictor selection, as this approach does not depend on the repeated measurement of the mental health-related variables.

- The integration of discussion into the results section is confusing. Please fully report the numerical results with appropriate statistics in the Results section and move the discussion to a separate Discussion section. It is especially important to completely report the Results from the model for the selected predictors.

We have carefully considered the feedback and acknowledge the concerns raised. However, we strongly believe that it is important to discuss the process of predictor selection before proceeding with the second analysis. This will help the readers understand the rationale behind the subsequent model generation and why certain predictors were chosen. If other reviewers agree with Reviewer#1, we will, of course, change the structure as requested.

- Line 111: Reduced after what treatment and over what time?

The methods section clarifies the treatment procedures. However, the journal's requirements mandate that the methods be structured at the end. To address this, we have added a comment in line 52 that directs readers to the methods section for information on treatment before proceeding to the main text.

- Table 3’s title is incorrect- these are not patient characteristics but the distributions of patient survey assessments at the two timepoints

This has been rephrased.

- The limitations of the study need to be more comprehensively stated

The limitations have been reviewed following Reviewer#2's suggestions.

CONCLUSIONS

- Greater precision is required regarding the statements about treatment (what treatment?), the time course of assessment, and the population examined.

Please see the methods section for all treatment-related information. Both treatment centers' therapeutic approaches are also detailed in published literature.

- **Line 222-224: This statement needs to be qualified as applying only to patients diagnosed with both tinnitus and depression. It is not clear to me that tinnitus should automatically be assessed in depressed patients without self-report or that this merits a revision to treatment guidelines.**

This has been rephrased.

METHODS

- **The treatments received by the cohorts need comprehensive description.**
- **The timepoints of assessment need to be linked with the use of T1 and T2 in the Results section. Differences in the duration of or type of therapy between cohorts need to be stated.**

The treatments (and their differences between centers) are now included in detail in the methods section; see response to Reviewer #2, while still referring to the existing publications regarding treatment regimens in the respective treatment centers. Time points have been clarified.

- **Please use more precise subheadings to better organize the information in the Methods.**

We divided the first section into two subsections and renamed the subheadings within "Analysis 2".

- **Please add that patients had confirmed clinical diagnosis of subjective, chronic tinnitus and the cutoff dates of diagnosis for inclusion**

This has been added.

- **Include dates of data collection for both cohorts**

This has been included.

- **Please state the minimally important clinical differences for all survey instruments used.**

We do not aim to compare treatment efficacy between centers, we consider differences in survey instruments advantageous to the overall quality and generalizability of our results. This allowed us to investigate whether the same predictors hold significance in tinnitus treatment, irrespective of the questionnaires used. Please also see our response to Reviewer#2 regarding treatment differences.

FIGURES

- **The captions of both figures need titles and to be explained in plain language. Move all abbreviation definitions to the end of the caption.**

To improve visibility, the caption titles for both figures have been highlighted in bold font. Abbreviations have been moved to the end of the caption. All statistical terms are explained in the main text.

Reviewer #2:

Remarks to the Author:

The study examines how tinnitus-related distress and depression interact during therapy, showing a significant two-way relationship. Somatic symptoms and perceived stress also play important roles. By analyzing large datasets from two treatment centers and using machine learning techniques, the research suggests that addressing tinnitus-related distress can improve the success of depression therapy, and vice versa. The article thoroughly explores this dynamic interaction, highlighting the roles of somatic symptoms and perceived stress.

However, to further improve the article's overall strength and accessibility, addressing specific aspects and enhancing communication is recommended.

In the “Main”, the flow between the overall discussion about distress caused by tinnitus and the specific emphasis on depressive symptoms could be improved. It would be beneficial to introduce the link between tinnitus and psychological distress earlier to prepare for the subsequent focus on depression.

The first paragraph of the introduction already discusses the link between tinnitus and psychological distress. Therefore, the abstract has been modified to provide additional clarification, preparing the reader for the psychological viewpoint of this work.

The introduction does not explicitly state the objective of the study. Although it is implied that the study aims to investigate the dynamic interaction between tinnitus-related distress and depression severity, it would be beneficial to clearly state this objective for clarity. For instance, consider adding a sentence towards the end of the introduction that summarizes the primary goal of the research.

This has been resolved. A sentence stating the primary objective more concisely has been added towards the end of the introduction.

The authors clarify that the data comes from two different centers (JUH and Charité Universitätsmedizin Berlin). However, they do not provide specific details on how they addressed or considered the differences between the two centers during the analysis. A more detailed explanation of how differences in methods or questionnaires might impact the results would have provided greater clarity.

Differences in treatment strategies are now thoroughly discussed in the methods section instead of simply referring to the literature introducing both treatments. Our approach takes into account the differences in questionnaires, treatment settings, and procedures, which we believe is advantageous. With a bicentric approach, we can examine whether the same predictors are significant in tinnitus treatment, independent of the treatment setting and the questionnaires utilized to measure change. This approach enables us to generate more accurate results and enhances the overall quality of the research.

Although the authors mention the use of the Elastic Net (ENet) model for predictor selection, they do not provide a detailed discussion of the limitations of this approach, especially about the high Root MSE mentioned in the criticisms. The authors state that the limited number of observations might compromise the accuracy of parameter estimates. However, they do not provide suggestions on how to address this limitation in detail or how it might impact the conclusions of the analysis. The authors fail to provide a comprehensive evaluation of the strengths and limitations of ENet, despite explaining its use. A more thorough discussion would have enhanced our understanding of the reliability of conclusions drawn from this method.

This has been resolved. The limitations section has been expanded to provide a better explanation of the high RMSE.

The interpretation of associations between variables is ambiguous, as the authors do not offer a clear explanation. A more detailed explanation of how to interpret these associations and their clinical implications would have improved our understanding of the results. The authors briefly mention predictor selection and model construction, but they do not provide specific details on the selection process or how these factors may influence the results. A more in-depth analysis of these aspects would have been beneficial.

We are not sure what the reviewer refers to here. Analysis 1, which covers several pages in the manuscript, is devoted to predictor selection. We try to explain the association between variables on

every level, as well as their direction and clinical significance. We would need more information from the reviewer in order to respond more properly.

Taking into account these considerations, if the authors address the points mentioned above, the work appears to be well-organized and well-written. The methods used are rigorous, and with careful revision, the work could potentially achieve even greater clarity.

This is a very encouraging remark; thank you!

We are grateful to the reviewers for taking the time to review our manuscript and for providing valuable comments and suggestions. We have carefully considered each suggestion and addressed it below. Due to the constructive input, we believe that the manuscript has further improved, enhancing the clarity of our work.

Reviewers' comments will be in bold font, while our responses will be in light font.

Reviewer #1 (Remarks to the Author):

The authors conducted a retrospective cohort study to examine the relationship between tinnitus-related distress and depression in two independent cohorts with tinnitus receiving therapy in Germany. They used a regression-based machine learning method to select predictors of treatment success using data from Cohort A (Jena University Hospital, n=500) which were applied to the data from Cohort B (Charite Universitätsmedizin Berlin, n=516) in a latent growth curve model. I am reviewing an updated version following revision to address comments from another journal. The authors have partially addressed my prior comments.

ABSTRACT

1. Regarding the term "daycare unit," please at least put "outpatient clinic" in parentheses to avoid confusion.

We understand that the term "daycare unit" can be confusing, so we have opted to use the term "day clinic" instead. We prefer not to use the term "outpatient clinic" as it does not fully capture the comprehensive care and treatments provided at both tinnitus centers. In Germany, an outpatient clinic typically provides only brief appointments and less intensive care, while a day clinic offers more comprehensive treatments and support for patients who spend most of their day at the clinic during their treatment.

2. As in my prior review of this paper, please add to the abstract:

- **The number of patients in the datasets and the selection criteria—did all patients have severe chronic subjective tinnitus?**
- **The years of data collection**
- **The global region of data collection**
- **That the therapy received was one of two forms of habituation therapy for severe chronic subjective tinnitus**
- **The duration of therapy in each of the two cohorts**
- **How distress and depression were assessed (what surveys)**
- **The numerical results of 'weaker' associations**
- **How sex was specifically related to the rate of change in depression**

The current abstract is 161 words, and the limit is 250, so there should be enough words to briefly describe the study details.

We have amended the abstract by incorporating the suggested information. After including all the suggested details, the word count slightly exceeded the limit, preventing submission, despite our efforts to keep the text as concise as possible. This means that we had to refrain from listing the names of the various depression surveys because the abbreviations could not be explained. If the editors and reviewers consider it essential, we are open to including all survey details. However, given the limited word count permitted, this may affect the inclusion of other important content in the abstract. All questionnaires are specified in detail in Table 1.

We have also removed the paragraph discussing anxiety and sex-related results from the abstract as it might distract the reader from the main point. Both results are extensively covered in the main text.

INTRODUCTION

1. It needs to be clear that the form of tinnitus being studied is subjective chronic tinnitus. Otherwise, general statements like in Line 27-28 (about curative treatment) are incorrect as other forms may be curable. If the form is more specific than that (severely bothersome) that should be clarified.

As the introduction already begins by defining subjective tinnitus, we have emphasized in the abstract and methods section that patients were treated for chronic *subjective* tinnitus. The statement regarding curative treatment has also been clarified.

2. Line 18-20: The authors should recognize the impact of geographical location on tinnitus incidence and prevalence, as recently summarized by Jarach et al. JAMA Neurol. 2022.

We have incorporated the influence of geographical biases on the variability in tinnitus prevalence estimates, citing Jarach et al., 2022.

3. Line 28: The guideline needs to be cited. This needs to be clarified that it is a German guideline. <https://doi.org/10.1007/s00106-022-01207-4>

This has been clarified, and the citation has been added.

4. Lines 50-52: While methodological details of the therapy received are fine to keep in the Methods section, it is important that the general type of treatment be mentioned in the Introduction, particularly given this article's placement of the Methods at the end and the stated aim of informing treatment strategies. Please add that the centers' treatment included two forms of tinnitus habitation therapy (or another descriptor). Similarly, add that these patients had clinically confirmed tinnitus-related distress.

The suggested information has been added to the introduction. The Methods section has been moved, as this journal allows for a different structure.

RESULTS/DISCUSSION

1. The article lacks a table including the demographic details of the two patient populations. Age and sex are summarized for the cohort in various text locations but the clarity of the article would benefit from including the baseline characteristics of the two populations in a table, or a dedicated first paragraph in the Results describing the two populations.

We have incorporated the baseline characteristics into the respective tables. Each results section already includes a dedicated paragraph detailing the distribution of age and sex in each sample. Following the feedback provided, the article has been reorganized such that the results are not interrupted by the discussion section.

2. The use of "t1" and "t2" would be better labeled in the tables with the exact timepoints assessed ("start of therapy" / "end of therapy" in the headings)

We have relabeled the time points as "pre" and "post" in all Tables and throughout the entire text.

3. I maintain that the integration of the Results and Discussion sections is confusing. If the process of predictor selection needs to be described first, then the Methods should be placed before the

Results for clarity. This journal accommodates that. A separate Discussion section would also allow the authors to better highlight the novelty of the study and break up large blocks of text with impede readability. The limitations should be placed in this section together, for all analyses.

The article has been restructured according to the reviewers' suggestions, taking into account the journal's style and formatting guide. The limitations section has been reorganized to include both analyses under one subheading.

4.

Reviewer #3 (Remarks to the Author):

In addition to a reading of the revised MS, this reviewer considered the comments from the original reviewer #2. One area in particular remains a concern, as the reviewer comment starting, "The interpretation of associations..." was not addressed by the authors. My interpretation of the comment would center, at least in part, on the authors providing a few examples from their analysis that could point to specific intervention considerations that could enhance the reader's understanding of the results, and perhaps increase the putative benefit for reader-clinicians. For example, the two sites employed different intervention doses; were there any site-specific elements of intervention that might have contributed to the difference observed regarding TQ score changes (ie., the JUH scores improved by about 2x those of Charite)? This reviewer can only speculate regarding the original reviewer's thinking; nevertheless, the suggestion above may merit consideration.

Thank you for this valuable comment. Both treatments are based on the same principles and mechanisms. However, two main idiosyncratic differences may have contributed to the observed difference in TQ score changes across treatment centers: 1. differences regarding the inclusion criteria and 2. the standardized use of hearing aids at JUH. Both differences have been addressed in the methods section for each treatment center.

1. Patients in the JUH sample experience more severe tinnitus-related distress pre-treatment, indicating the statistical/ numerical potential for greater improvement compared to Charité.
2. At JUH, we provide each patient with individually fitted hearing aids using a standardized setting. When considering the isolated treatment effect of the hearing aids as reported by Boecking et al., 2022 (<https://doi.org/10.3390/jcm11071764>), along with the counseling effect, it is evident that JUH surpasses the expected treatment changes. We hypothesize that there is a super-additive interaction between both treatment components. However, due to the speculative nature of this hypothesis, we chose not to include it in the article.

While we consider this topic of interest for future research, we believe it may distract from the main point of our current paper. Our goal is not to compare treatment regimens but rather to utilize the bicentric design and site-specific differences to validate the significance of predictors cross-contextually while aiming to explore the interaction between tinnitus-related distress and depression severity across treatment. However, we are open to including this topic in the discussion if recommended by the reviewer.

Main:

Consider rewording a few passages.

Line 27: consider wording change to, "...psychotherapeutic interventions, and sound-based interventions aimed at alleviating tinnitus-related distress as opposed to eradicating the tinnitus sound."

Lines 31-32: consider wording change to, "...and related comorbidity hinders attempts at predicting

tinnitus-related distress and treatment response [8]. Additionally, the application....”

Lines 36-37: consider wording change to, “...psychological features may be more strongly associated with tinnitus severity than measures purporting to characterize the tinnitus sound.”

Line 66: Would “interdisciplinary” be more appropriate than “transcontextual”?

We have implemented all the above suggestions. In using the term transcontextual, we intended to convey the idea of the identified predictors maintaining their relevance across different contexts (i.e., therapy settings) in order to understand their broader applicability. For clarity, we have replaced “transcontextual” with “cross-contextual” and revised the paragraph.

Results/Discussion:

This reviewer agrees with reviewer 1’s comments from the original MS in that the shifting from results to discussion within each section is a bit confusing. I understand the authors’ rationale, however the current structure impedes somewhat the flow of the article.

The article has been restructured according to the reviewers’ suggestions, taking into account the journal’s style and formatting guide.

Line 73 (and elsewhere): The authors need to report in either the methods or results sections the 12-point criterion change on the TQ required for clinical significance. Their reported 14.5 point change exceeds the criterion value, however it would be of greater value to the reader if the percentage of participants whose scores exceeded the criterion value were also reported. For example, an average of 14.5 change COULD mean that everyone in the study experienced significant improvement, even as some participants remained in the decompensated condition. It might also be true (particularly at Charite) that participants switched from decompensated to compensated even if their score change did not exceed the criterion. Tables 2&3 provide the reader with the number of participants whose score changes resulted in “compensated” tinnitus, but the reader does not know how many participants exceeded the criterion value for improvement. Perhaps the authors could comment on the number of participants whose classification changed from decompensated to compensated, but whose TQ score change did not exceed 12 points. At the very least, the authors could report % of participants whose TQ scores showed significant improvement.

We calculated the percentages of participants whose scores exceeded the criterion value for both treatment centers and added the information to the respective paragraphs in the results section, citing Hall et al., 2018.

We also calculated the number of participants whose classification changed from decompensated to compensated without exceeding the 12-point improvement. In the JUH sample, 3.8% (N=19) and in the Charité sample, 4.04% (N=41) experienced an improvement from decompensated to compensated symptom severity despite a change score below 12 points. Due to the small number of patients who fall into this category, we opted not to report these values in the main text.

Conclusion:

Line 250: this reviewer prefers “symptom” to “syndrome” when labeling tinnitus, as the latter connotes phenotypic presentations that may be clinically validated in ways that tinnitus cannot.

The wording has been corrected. The authors completely agree with this perspective. Thank you for bringing this mistake to our attention.

Reviewer #4 (Remarks to the Author):

The authors investigated the relationship between tinnitus-related distress and depression and

their bi-directional relationship. The data presented was collected from two sites, with data from one site used for predictor selection. The predictors were then used with data from the second site to investigate changes in tinnitus-related distress and depression at two timepoints during therapy. Findings show a bidirectional association between tinnitus-related distress and depression severity, across treatment.

The existence of distress and depression in many patients with tinnitus is known, and the novelty of this study appears to be establishing the bidirectional relationship. I believe the contribution should be clarified to better acknowledge previous knowledge in the field.

We have reiterated the aim of our study and the aspect of novelty in this finding throughout the introduction, the discussion, and the conclusion.

As the study did not aim to assess efficacy of treatments as a main outcome, the last line of the abstract could be edited to clarify that “enhanced therapy success in depression when tinnitus-related distress is addressed and vice versa”, is proposed for future treatment strategies (as in this study there was no comparison of different treatment strategies).

This has been clarified.

When selecting predictors (analysis 1), the authors include anxiety and exclude lateralization, contrary to what the predictor selection shows. This decision appears somewhat arbitrary, and it would seem more reasonable, in terms of the analysis methods, to report results without including/ removing the mentioned factors.

The authors agree with this comment as the decision does indeed appear contradictory. Initially, we had intended to conduct an unbiased analysis of the first dataset to select predictors. However, not all predictors that were examined in one dataset were available in the second dataset, including tinnitus lateralization. Based on the evidence from post-selection adjusted confidence intervals in a previous study (Ivansic et al., 2022), it seems that the potential impact of tinnitus lateralization on treatment outcomes is likely not a true association with tinnitus-related distress. Accordingly, we have removed this variable entirely from the analyses and re-run the ENet without this predictor to improve clarity.

We have decided to uphold the choice to include anxiety in the second analysis in order to re-evaluate previous doubts about its true connection with tinnitus-related distress. By using datasets from two different treatment sites, this approach enables us to validate predictors across different settings. Furthermore, the model in analysis 2 offers an opportunity to gain insight into the mechanisms why anxiety might not emerge as a predictor. We believe this objective is of interest to the research community and, therefore, propose to retain the mentioned factor in our subsequent analysis.

Line 76: The authors mention ENet resulted in features in chronological order. It is not clear what they mean by chronological in this context.

This has been reworded for clarity.

In the description for Table 1, they highlight that scores were transformed to a common scale. How is the transformation done?

This information has been added to the description of Table 1.

It is unclear where some of the results are shown. For example, line 91 mentions the positive regression coefficient of the TQ score at t1 but it is unclear where the coefficients are reported.

We have included a table in the supplementary part disclosing all regression coefficients. As the coefficients are not directly interpretable due to the impact of the penalty term, we only interpret the comparative strength of associations and the prediction direction in the main text. We have inserted a note referring the reader to the supplementary part.

The authors kindly thank all reviewers and the editor for reviewing our manuscript. We have carefully considered the remaining suggestions and addressed them below. Reviewers' comments will be in bold font, while our responses will be in light font.

REVIEWERS' COMMENTS:

Reviewer #1 (Remarks to the Author):

The authors have largely responded to my prior comments, which has enhanced the clarity of the manuscript. For the newly organized Discussion section, it would benefit from a more traditional article format such as a starting paragraph which summarized the main findings before delving into the detailed interpretation of the two analyses. Breaking up long paragraphs can also help with the ease of reading of that section. My lingering criticism is the lack of numerical/precise reporting of the results in the Abstract, which instead uses the majority of the word count to make general statements. If this is typical for the journal's format, that's fine. Publication is recommended.

Thank you for your valuable comments and for reviewing our work. Due to the editor's recommendation, we have retained the abstract in the present format. In the discussion section, we have added a paragraph summarizing the main findings at the beginning of the discussion. We have also broken up the text into several paragraphs and retained the subheadings, as we agree that this immensely improves readability. However, upon the editor's request, we are open to removing the subheadings to comply with the formatting guidelines.

Reviewer #3 (Remarks to the Author):

The authors are to be commended for addressing reviewer comments, and in particular, comments from a newly-recruited reviewer. It is expected that this MS will be of interest to the readership.

Thank you for reviewing our work.

Reviewer #4 (Remarks to the Author):

Thank you to the authors. My comments have been addressed.

Thank you for reviewing our work.